# Diagnosis of Non-Alcoholic Fatty Liver Disease (NAFLD) Is Independently Associated with Cardiovascular Risk in a Large Austrian Screening Cohort

**DOI:** 10.3390/jcm9041065

**Published:** 2020-04-09

**Authors:** David Niederseer, Sarah Wernly, Sebastian Bachmayer, Bernhard Wernly, Adam Bakula, Ursula Huber-Schönauer, Georg Semmler, Christian Schmied, Elmar Aigner, Christian Datz

**Affiliations:** 1Department of Cardiology, University Heart Center Zurich, University of Zurich, University Hospital Zurich, 8091 Zurich, Switzerland; adam.bakula@usz.ch (A.B.); christian.schmied@usz.ch (C.S.); 2Department of Internal Medicine, General Hospital Oberndorf, Teaching Hospital of the Paracelsus Medical University Salzburg, 5110 Oberndorf, Austria; sarah_wernly@airpost.net (S.W.); S.Bachmayer@kh-oberndorf.at (S.B.); huber.schoenauer@gmail.com (U.H.-S.); georg.semmler@hotmail.com (G.S.); c.datz@kh-oberndorf.at (C.D.); 3Department of Internal Medicine II, Paracelsus Medical University Salzburg, 5020 Salzburg, Austria; bernhard@wernly.net; 4Department of Internal Medicine I, Paracelsus Medical University Salzburg, 5020 Salzburg, Austria; e.aigner@salk.at

**Keywords:** NAFLD, cardiovascular risk, Framingham risk score, CVD, risk prediction, secondary prevention, primary prevention, metabolic syndrome, NAFLD fibrosis score

## Abstract

Background: Many patients with non-alcoholic fatty liver disease (NAFLD) simultaneously suffer from cardiovascular (CV) disease and often carry multiple CV risk factors. Several CV risk factors are known to drive the progression of fibrosis in patients with NAFLD. Objectives: To investigate whether an established CV risk score, the Framingham risk score (FRS), is associated with the diagnosis of NAFLD and the degree of fibrosis in an Austrian screening cohort for colorectal cancer. Material and Methods: In total, 1965 asymptomatic subjects (59 ± 10 years, 52% females, BMI 27.2 ± 4.9 kg/m^2^) were included in this study. The diagnosis of NAFLD was present if (1) significantly increased echogenicity in relation to the renal parenchyma was present in ultrasound and (2) viral, autoimmune or hereditary liver disease and excess alcohol consumption were excluded. The FRS (ten-year risk of coronary heart disease) and NAFLD Fibrosis Score (NFS) were calculated for all patients. High CV risk was defined as the highest FRS quartile (>10%). Both univariable and multivariable logistic regression models were used to calculate associations of FRS with NAFLD and NFS. Results: Compared to patients without NAFLD (*n* = 990), patients with NAFLD (*n* = 975) were older (60 ± 9 vs. 58 ± 10 years; *p* < 0.001), had higher BMI (29.6 ± 4.9 vs. 24.9 ± 3.6 kg/m^2^; *p* < 0.001) and suffered from metabolic syndrome more frequently (33% vs. 7%; *p* < 0.001). Cardiovascular risk as assessed by FRS was higher in the NAFLD-group (8.7 ± 6.4 vs. 5.4 ± 5.2%; *p* < 0.001). A one-percentage-point increase of FRS was independently associated with NAFLD (OR 1.04, 95%CI 1.02–1.07; *p* < 0.001) after correction for relevant confounders in multivariable logistic regression. In patients with NAFLD, NFS correlated with FRS (*r* = 0.29; *p* < 0.001), and FRS was highest in patients with significant fibrosis (F3-4; 11.7 ± 5.4) compared to patients with intermediate results (10.9 ± 6.3) and those in which advanced fibrosis could be ruled-out (F0-2, 7.8 ± 5.9, *p* < 0.001). A one-point-increase of NFS was an independent predictor of high-risk FRS after correction for sex, age, and concomitant diagnosis of metabolic syndrome (OR 1.30, 95%CI 1.09–1.54; *p* = 0.003). Conclusion: The presence of NAFLD might independently improve prediction of long-term risk for CV disease and the diagnosis of NAFLD might be a clinically relevant piece in the puzzle of predicting long-term CV outcomes. Due to the significant overlap of advanced NAFLD and high CV risk, aggressive treatment of established CV risk factors could improve prognosis in these patients.

## 1. Introduction

With a constant increase in the incidence of metabolic syndrome, the prevalence of non-alcoholic fatty liver disease (NAFLD) is estimated to be around 25% in Europe. A steep rise in the prevalence of NAFLD from 15% in 2005 to 25% in 2010 has been observed [1]. This increase mirrors obesity rates, which nearly tripled since 1975 and reached epidemic levels [2]. Components of the metabolic syndrome such as hypertension, dyslipidemia, dysglycemia, and abdominal obesity are established risk factors for NAFLD [3]. Since they have also been established as risk factors for CVD, patients frequently suffer from both conditions. 

CVD is a leading cause of death worldwide both in the general population and patients with NAFLD [4,5,6]. NAFLD is independently associated with several markers of subclinical atherosclerosis such as coronary artery calcification, impaired flow-mediated vasodilation, arterial stiffness, carotid artery inflammation and thickening of carotid intima-media as well as left ventricular hypertrophy and diastolic dysfunction [7,8]. Importantly, some of these studies suggest an association of these two disease entities independent from traditional risk factors. Several lines of evidence suggest that NAFLD may be causally and independently involved in CVD pathogenesis [9,10]. 

Different possible pathophysiological pathways link NAFLD with CVD [11]. Markers of inflammation such as cytokines, CRP, or interleukin-6 are overexpressed in these patients and also correlate with a higher degree of liver fibrosis [12]. Furthermore, patients with hepatic steatosis show elevated levels of pro-coagulant factors such as fibrinogen, von Willebrand factor and plasminogen activator inhibitor-1 [13]. Additionally, hepatic insulin resistance and atherogenic dyslipidemia seem to contribute to the development of CVD [14]. These mechanisms are possible explanations for the fact that the severity of NAFLD, especially if progressed to non-alcoholic steatohepatitis (NASH) with fibrosis, additionally contributes to CV risk [15]. 

In our study, we examined the prevalence of NAFLD in an Austrian screening cohort for colorectal cancer (SAKKOPI). An established non-invasive estimate of fibrosis severity i.e., the NAFLD fibrosis score (NFS) was calculated and the relation of fibrosis with CV risk as assessed by the Framingham Risk Score (FRS) evaluated.

## 2. Methods

### 2.1. Study Subjects

The study cohort consisted of 1965 Caucasians undergoing routine screening colonoscopy at a single center in Austria. All Patients were recruited between 2010 and 2014. Informed consent was obtained, and the study was approved by the local ethics committee (Ethikkommission des Landes Salzburg, approval no. 415-E/1262/2-2010). 

### 2.2. Assessment

As previously described, participants were examined on two consecutive days [16]. On the day of admission, venous blood was drawn after an overnight fast. A whole blood count, kidney and liver tests, lipids, CRP, as well as hemoglobin A1c, an oral glucose tolerance test, and insulin levels were measured. The participants completed a detailed questionnaire including past medical history, current medical regimen, family history, smoking history (“never smokers”, “former smokers”, or “current smokers”) dietary habits and physical activity. A standard physical examination including blood pressure, height, weight, and waist circumference) was performed. Importantly, all patients underwent abdominal ultrasonography. The liver was considered normal if echogenicity was similar to the renal parenchyma. If areas showed a significantly increased echogenicity compared to the renal parenchyma, the liver was considered steatotic. On the second day, all subjects underwent complete colonoscopy.

### 2.3. Definitions

The diagnosis of NAFLD was made after exclusion of viral, autoimmune and hereditary liver diseases (Wilson disease, hereditary haemochromatosis, alpha-1 antitrypsin deficiency) and excess daily alcohol consumption ≥30 g for men and ≥20 g for women according to the European clinical practice guidelines for the management of NAFLD [17]. NAFLD fibrosis score (NFS) was calculated as previously described [18]. Briefly, NFS (age, body mass index (BMI), presence of impaired fasting glucose or diabetes, aspartate-aminotransferase (AST), alanine-aminotransferase (AST), platelets and albumin) was used to stratify patients according to their risk of significant fibrosis. Specifically, patients with a NFS < −1.455 were graded as F0-2, those with NFS > 0.676 as “F3-4”, and patients with a NFS between −1.455 and 0.676 as “intermediate”.

Metabolic syndrome was diagnosed when three or more of the following criteria were met [19]: fasting blood glucose level ≥100 mg/dL or antidiabetic therapy, waist circumference >102 cm in males and >88 cm in females, blood pressure ≥130/85 mmHg or current antihypertensive treatment, plasma triglycerides ≥150 mg/dL, and plasma HDL <40 mg/dL in males and <50 mg/dL in females. 

### 2.4. Cardiovascular Risk Assessment

We evaluated patients for cardiovascular disease applying the Framingham Risk Score (FRS) [20]. Although the FRS is not validated in subjects with diabetes (T2DM), we did include subjects with T2DM in our analysis and performed a separate analysis, excluding all subjects with T2DM. Since results were not changed when subjects with T2DM were excluded, we report the results including T2DM to allow for greater generalizability of our results.

### 2.5. Statistical Analysis

Continuous variables are expressed as mean (±standard deviation) and compared using *t*-test or ANOVA. Categorical data are expressed as numbers (percentage). Chi-square test was applied to calculate differences between groups. Both univariable and multivariable logistic regression was used to evaluate associations of FRS with NAFLD and NFS with CV risk. For multivariable logistic regression, elimination criteria was a *p*-value of < 0.10 following backward elimination. Variables were included in the multivariable model based on literature. All variables included in the multivariable models evidenced a univariable association at a *p*-value of *p* < 0.05. A *p*-value of < 0.05 was considered statistically significant. SPSS version 22.0 (IBM, USA) was used for statistical analyses. 

## 3. Results

### 3.1. Analysis of the Total Study Cohort, NAFLD versus Non-NAFLD Patients

Overall, 49.6% (*n* = 975) of patients had NAFLD as defined by hepatic steatosis in ultrasound, while 990 patients (50.4%) did not have NAFLD. NAFLD patients were older (60 ± 9 vs. 58 ± 10 years; *p* < 0.001), evidenced higher BMI (29.6 ± 4.9 vs. 24.9 ± 3.6 kg/m^2^; *p* < 0.001) and more frequently fulfilled criteria for metabolic syndrome (33% vs. 7%; *p* < 0.001). Characteristics of NAFLD versus non-NAFLD patients are shown in Table 1.

CV risk assessed by FRS was higher in the NAFLD-group (8.7 ± 6.4 vs. 5.4 ± 5.2%; *p* < 0.001). After allocation of subjects to FRS into risk quartiles (Q1: FRS 0%–2%; Q2: FRS 2%–5%; Q3: FRS 5%–10%, Q4: FRS > 10%), patients with NAFLD more often were in the Q4-FRS group (33% vs. 16%; *p* < 0.001) compared to non-NAFLD patients. 

In univariable logistic regression, this relationship corresponded to an increase of OR of 1.11, (95%CI 1.09–1.13; *p* < 0.001) in the likelihood for NAFLD per one-percentage-point increase of FRS. This association remained significant after correction for age, sex and metabolic syndrome (OR, 1.04 95%CI 1.02–1.07; *p* < 0.001) in a multivariable model (Table 2). In an additional sensitivity analysis, a one-percentage-point increase of FRS remained associated with an increased likelihood for NAFLD both in males (OR 1.08, 95%CI 1.06–1.11; *p* < 0.001) and females (OR 1.13, 95%CI 1.09–1.18; *p* < 0.001).

### 3.2. Analysis of Patients with NAFLD

Patients with NAFLD were grouped according to their NFS into F0-F2 (*n* = 604), intermediate (*n* = 138) and F3-4 (*n* = 10). The characteristics of patients according to their NFS are shown in Table 3. Over the whole NAFLD cohort, NFS correlated with FRS (*r* = 0.29; *p* < 0.001), and FRS was highest in the F3-4 group (11.7 ± 5.4%; *p* < 0.001 vs. F0-F2) compared to the intermediate (10.9 ± 6.3%) and the F0-F2 group (7.8 ± 5.9%). When grouping intermediate and F3-4 into an “at-risk” group (due to small sample size in F3-4), the significant differences between F0-2 essentially persisted (Table 4).

In univariable logistic regression, a one-point increase of NFS was associated with a higher likelihood of high-risk FRS (OR 1.60, 95%CI 1.41–1.83; *p* < 0.001). NFS remained an independent predictor of Q4-FRS after correction for sex, age, and concomitant diagnosis of metabolic syndrome (OR 1.30, 95%CI 1.09–1.54; *p* = 0.003). In a sensitivity analysis in both males (OR 1.84, 95%CI 1.54–2.20; *p* < 0.001) and females (OR 2.06, 95%CI 1.52–2.78; *p* < 0.001) a one-point increase of NFS remained associated with high-quartile FRS. Univariable and multivariable significant associations of age, female gender, metabolic syndrome, and FRS with the presence of high risk NFS are depicted in Table 5.

## 4. Discussion

Our study confirms that there is a “silent epidemic” of NAFLD. In the present cohort of asymptomatic individuals undergoing colonoscopy screening between 50 and 75 years of age, around 50% were diagnosed with NAFLD. In total, 14.2% of the screened patients were categorized as being intermediate and 1% of patients were at high risk for advanced fibrosis by the NFS. Importantly, patients with NAFLD had higher CV risk as defined by the FRS compared to patients without NAFLD. Finally, the CV risk was highest in patients with highest NFS scores. 

The NFS does not only predict the risk for advanced liver fibrosis, but also CV risk. Interestingly, in a post-hoc analysis of the IMPROVE-IT trial the NFS identified patients who were at the highest risk for recurrent cardiovascular events. The IMPROVE-IT compared statin therapy alone to the add-on of ezetimibe in post ACS patients [21]. In this trial, higher NFS identified patients more likely to benefit from aggressive lipid-lowering therapy. Thus, although the IMPROVE-IT trial was not designed to assess the link between NAFLD and ACS, it offers important data on the potential link between fatty liver severity and atherosclerosis [21]. 

The most obvious link between CV risk and NFS is the fact that this score is constituted of factors like age, BMI, ALT, AST, platelets, albumin, and the presence or absence of diabetes, all of which reflect metabolic and inflammatory processes. Of note, inflammation and fibrosis are hallmarks of both liver and cardiovascular disease [18] and may therefore indicate common systemic mechanisms.

In our analysis, NAFLD was an independent risk indicator for CV risk. This is in concordance with a meta-analysis of pooled studies from European, Asian, and American countries suggesting an independent association of NAFLD with CV risk [10]. However, a British study including 17.7 million patients found that the diagnosis of NAFLD was not associated with increased risk for acute myocardial infarction or stroke after adjustment for established CV risk factors [22]. Nevertheless, in another meta-analysis of Targher et al., patients with NAFLD evidenced an increased risk of fatal and non-fatal CV disease [23]. Although the link between NAFLD and CV risk seems intuitive, the effect on CV mortality or events has not been demonstrated. Also, a role for a specific medical treatment for NAFLD in preventing CV events and mortality beyond lifestyle advice and current CV guidelines is not established [24,25]. The data in this manuscript suggests an independent relationship of CV risk and NAFLD in an Austrian cohort. Specific management strategies may be considered based on this evidence to improve liver outcomes in CV patients and CV outcomes in liver patients.

### 4.1. CV Risk Assessment for NALFD Patients

Considering the Joint Clinical Practice Guidelines of EASL-EASD-EASO for the management of NAFLD patients [17], a non-invasive test should be used as the first screening tool to assess disease severity. Depending on the result, patients can be graded into low, intermediate and high risk with regard to advanced fibrosis. For patients in the low risk group, their individual cardiovascular risk should be assessed by risk scores as for example by the FRS. Target goals for risk factors, e.g., for blood pressure, LDL levels, body weight or blood glucose should be treated according to primary prevention guidelines [25]. 

Patients with intermediate and high risk for advanced fibrosis should be referred to a hepatologist. In patients with advanced fibrosis stage or even cirrhosis CV risk should be assessed by a cardiologist as described in by Choudhary and Duseja [26]. All other patients should be clinically assessed, stratified by a CV risk score and should be managed according to respective prevention guidelines [25] (Figure 1).

### 4.2. Screening for NAFLD in CV Patients

For patients after an CV event or at with a high CV risk we suggest the following approach to detect NAFLD. As a screening test the NFS could be calculated. For patients with low risk for advanced fibrosis, lifestyle modification changes could be recommended. Patients with an intermediate risk could be referred to a liver ultrasound exam and to a hepatologist with expertise in transient elastography. If these exams show no fibrosis or a low stage of fibrosis they should be managed as patients in the low risk group. For patients with intermediate risk in the NFS and advanced fibrosis or cirrhosis in the further exams as well as for patients with a high risk NFS score a hepatologist should be consulted. We are aware, that NFS was developed to estimate fibrosis in the presence of NAFLD. However, we here propose NFS as cheap and non-invasive “screening tool” for NAFLD in patients after an CV event or with a high CV risk. All patients should be treated according to the current guidelines of the European Society of Cardiology [24] (Figure 2).

## 5. Limitations

This study is a post-hoc analysis of a single-center prospective register and the results remain thesis-generating. However, these data mirror a real-world Austrian population and indicate a high prevalence of undetected NAFLD in the general population. Although this study cannot provide longitudinal CV outcome data, we provide data from a carefully characterized cohort in a cross-sectional study. Another limitation of this study is the linearity of the models especially in using a high number of contributing factors, an assumption that is implicit due to the design of the study. 

Furthermore, ultrasound and not liver transient elastography was used to diagnose NAFLD. Finally, clinical data and established surrogate risk scores for calculation of the CV risk as well as the determination of the degree of liver fibrosis by non-invasive scores were used, even though there are other but more expensive and sometime even more invasive methods available to determine CV risk or liver fibroses such as magnetic resonance imaging, liver biopsy, liver transient elastography, vascular ultrasound, or coronary calcium scoring.

## 6. Conclusions

The presence of NAFLD might independently predict long-term risk for CV disease. Therefore, patients with high risk for or known CV events should be screened for the presence of NAFLD and risk scores should be routinely applied. Non-invasive risk scores for CV risk and fibrosis could help to facilitate and optimize management of patients with NAFLD with increased CV risk. The care for patients with both NAFLD and CV disease is challenging and due to the vast overlap of patients screening for liver disease in CV patients as well as screening for NAFLD in CV patients seems reasonable [26]. Cardiologists and hepatologists should team up in the treatment of their patients [27].

## Figures and Tables

**Figure 1 jcm-09-01065-f001:**
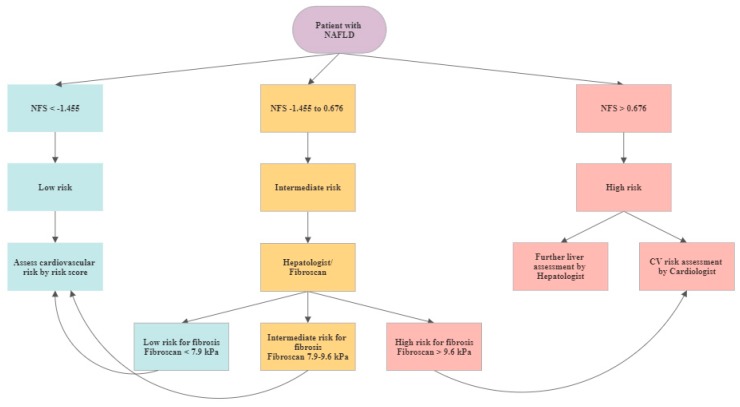
Cardiovascular (CV) assessment algorithm in patients with diagnosed NAFLD.

**Figure 2 jcm-09-01065-f002:**
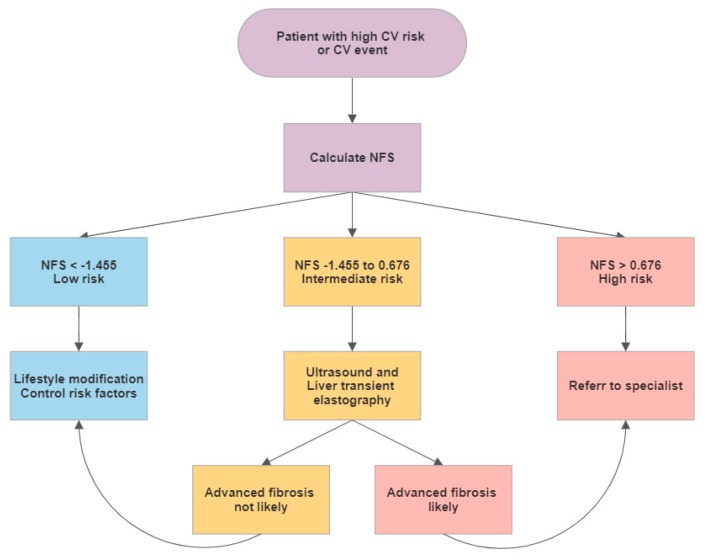
Liver assessment in patients with high cardiovascular risk or with a cardiovascular event in the past medical history.

**Table 1 jcm-09-01065-t001:** Baseline characteristics of patients without (*n* = 990) and with (*n* = 975) non-alcoholic fatty liver disease (NAFLD).

	No NAFLD	NAFLD	Total Cohort	*p*-Value
	*n* = 990	*n* = 975	*n* = 1965	
Female	61%	43%	52%	<0.001
Age (years)	58 (10)	60 (9)	59 (10)	<0.001
Systolic RR (mmHg)	128 (18)	135 (19)	131 (18)	<0.001
Diastolic RR (mmhg)	79 (10)	83 (11)	81 (10)	<0.001
BMI (kg/m^2^)	25 (4)	26 (5)	27 (4)	<0.001
Waist circumference (cm)	90 (11)	105 (12)	97 (11)	<0.001
Waist to hip ratio	1 (0.1)	1 (0.1)	1 (0.1)	<0.001
Bilirubine (mg/dL)	0.72 (0.4)	0.73 (0.4)	0.72 (0.4)	0.4
GGT (U/L)	31 (46)	48 (71)	40 (46)	<0.001
AST (U/L)	22 (12)	26 (18)	24 (12)	<0.001
INR	1.0 (0.1)	1.0 (0.1)	1.0 (0.1)	0.24
Total cholesterol (mg/dL)	219 (40)	217 (44)	218 (40)	0.25
HDL (mg/dL)	67 (18)	56 (16)	62 (18)	<0.001
LDL (mg/dL)	137 (36)	142 (39)	139 (36)	0.02
Triglycerices (mg/dL)	101 (51)	145 (85)	123 (51)	<0.001
Thrombocytes (G/L)	236 (66)	227 (65)	232 (66)	0.001
Fasting glucose (mg/dL)	97 (15)	109 (30)	103 (15)	<0.001
HbA1c (%)	5.6 (0.5)	5.9 (0.8)	5.8 (0.5)	<0.001
Metabolic syndrome	7%	33%	20%	<0.001
T2DM	9%	24%	16%	<0.001
Current smoker	19%	17%	20%	0.48
**Medication**				
ASS	11%	17%	14%	0.001
Statin	15%	23%	19%	<0.001
ACE-I/ARB	13%	27%	20%	<0.001
Metformin	2%	8%	5%	<0.001
**CV risk score**				
FRS	5.41 (5.20)	8.71 (6.38)	7.05 (5.20)	<0.001
FRS 0-2%	41%	19%	30%	<0.001
FRS >2–5%	21%	19%	20%	
FRS >5–10%	22%	30%	25%	
FRS >10%	16%	33%	24%	

NAFLD: Non-alcoholic fatty liver disease; NFS: NAFLD fibrosis score; FRS: Framingham Risk Score; RR: blood pressure; GGT: gamma-glutamyl-transferase; AST: Aspartate transaminase; INR: International normalized ratio; HDL: High-density lipoprotein; LDL: Low-density lipoprotein; HbA1c: Glycated hemoglobin; T2DM: type 2 diabetes mellitus; ASS: acetylsalicylic acid; CV: cardiovascular; OR: odds ratio.

**Table 2 jcm-09-01065-t002:** Univariable and multivariable associations with the presence of NAFLD.

		Univariable			Multivariable	
	OR	95%CI	*p*-Value	OR	95%CI	*p*-Value
Age	1.03	1.02–1.04	<0.001	1.010	0.998–1.023	0.11
Female gender	0.48	0.40–0.58	<0.001	0.68	0.54–0.86	0.001
Metabolic syndrome	6.08	4.63–7.99	<0.001	5.02	3.77–6.70	<0.001
FRS	1.11	1.09–1.13	<0.001	1.06	1.04–1.08	<0.001

**Table 3 jcm-09-01065-t003:** Baseline characteristics of patients according to their NAFLD Fibrosis Score (NFS) score: F0-F2 (*n* = 604), intermediate (*n* = 138) and F3-F4 (*n* = 10).

	F0-F2		Intermediate		F3-F4		
	*n* = 604		*n* = 138		*n* = 10		
	Mean	SD	Mean	SD	Mean	SD	*p*-Value
Female	36%		43%		50%		0.80
Age (years)	59	9	66	8	67	9	<0.001
Systolic RR (mmHg)	134	18	139	19	148	26	<0.001
Diastolic RR (mmhg)	82	11	85	12	85	12	0.07
BMI (kg/m^2^)	29	4	33	6	35	4	<0.001
Waist circumference (cm)	103	11	111	12	115	14	<0.001
Waist to hip ratio	0.96	0	0.97	0	0.97	0	0.23
Bilirubine (mg/dL)	0.70	0	0.80	1	1.57	1	<0.001
GGT (U/L)	48	76	53	70	115	145	0.02
AST (U/L)	25	15	30	24	55	61	<0.001
INR	0.99	0	1.02	0	1.17	0	<0.001
Total cholesterol (mg/dL)	221	44	202	42	221	52	<0.001
HDL (mg/dL	57	16	53	13	57	13	0.03
LDL (mg/dL)	145	40	130	37	142	41	<0.001
Triglycerices (mg/dL)	145	84	147	101	142	68	0.97
Thrombocytes (G/L)	243	62	176	52	128	88	<0.001
Fasting glucose (mg/dL)	107	28	115	28	97	16	0.01
HbA1c (%)	5.9	1	6.0	1	5.6	0	0.08
Metabolic syndrome	30%		43%		40%		0.01
T2DM	20%		44%		20%		<0.001
Current Smoker	19%		6%		0%		0.02
Medication							
ASS	24%		31%		13%		0.21
Statin	24%		31%		13%		0.21
ACE-I/ARB	22%		38%		20%		0.02
Metformin	8%		10%		0%		0.42
FRS	7.83	5.92	10.87	6.29	11.70	5.44	<0.001

**Table 4 jcm-09-01065-t004:** Baseline characteristics of patients according to their NFS score: F0-F2 (*n* = 604), and intermediate or F3-F4 (*n* = 148).

	F0-F2	Intermediate or F3-F4	
	*n* = 604	*n* = 148	*p*-Value
Female	41%	43%	0.64
Age (years)	59 (9)	66 (9)	<0.001
Systolic RR (mmHg)	134 (18)	140 (18)	<0.001
Diastolic RR (mmhg)	82 (11)	85 (11)	0.02
BMI (kg/m^2^)	29 (4)	33 (4)	<0.001
Waist circumference (cm)	103 (11)	111 (11)	<0.001
Waist to hip ratio	1 (0)	1 (0)	0.09
Bilirubine (mg/dL)	1 (0)	1 (0)	<0.001
GGT (U/L)	48 (76)	57 (76)	0.16
AST (U/L)	25 (15)	32 (15)	<0.001
INR	0.99 (0.07)	1.03 (0.07)	<0.001
Total cholesterol (mg/dL)	221 (44)	203 (44)	<0.001
HDL (mg/dL	57 (16)	53 (16)	0.01
LDL (mg/dL)	145 (40)	131 (40)	<0.001
Triglycerices (mg/dL)	145 (84)	147 (84)	0.87
Thrombocytes (G/L)	243 (62)	173 (62)	<0.001
Fasting glucose (mg/dL)	107 (28)	113 (28)	0.01
HbA1c (%)	5.9 (0.7)	6.0 (0.7)	0.12
Metabolic syndrome	30%	43%	0.003
T2DM	20%	40%	<0.001
Current Smoker	19%	6%	0.003
Medication			
ASS	16%	21%	0.11
Statin	24%	30%	0.20
ACE-I/ARB	22%	37%	0.01
Metformin	8%	10%	0.39
FRS	7.83 (5.92)	10.92 (5.92)	<0.001

**Table 5 jcm-09-01065-t005:** Univariable and multivariable associations with the presence of high risk NFS score.

		Univariable			Multivariable	
	OR	95%CI	*p*-Value	OR	95%CI	*p*-Value
Age	1.11	1.09–1.13	<0.001	1.17	1.14–1.21	<0.001
Female gender	0.15	0.10–0.21	<0.001	0.02	0.01–0.04	<0.001
Metabolic syndrome	2.46	1.86–3.26	<0.001	4.15	2.64–6.55	<0.001
FRS	1.60	1.41–1.83	<0.001	1.30	1.09–1.54	0.003

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
