# Peer review of "Diagnosis of Non-Alcoholic Fatty Liver Disease (NAFLD) Is Independently Associated with Cardiovascular Risk in a Large Austrian Screening Cohort"

_jcm, 2020, doi:10.3390/jcm9041065_

Round 1

Reviewer 1 Report

The paper by D.Niederseer et al is a well designed and executed clinical study in which the conclusions are well supported by the obtained results. The major concern of this reviewer deals with the interest of the paper for the readers. Although the cohort is large and well defined, there are other reported studies including largest group of patients obtaining quite similar conclusions. In summary, in my opinion the study is well performed but lacks originality.   

Author Response

Re: We thank Reviewer #1 for these encouraging remarks. We agree that similar findings have been reported previously. Some of these previous reports are discussed in the discussion section. Together with Reviewer #3 that “this study thus contributes nicely to an increasing pool of studies that define NAFLD as a “silent epidemic” as the authors write, and as a contributor to CVD.” We hope that with the multitude of reports on this evolving clinical problem also the awareness of clinicians will increase.

Reviewer 2 Report

This article is a post-hoc analysis of a patient cohort. The authors have correlated NAFLD with Framingham Risk Score. Their results are original and interesting. The manuscript is clearly written in a straightforward way. 

The fact that none clinical endpoints but a surrogate risk score has been utilised should be addressed in the limitations section.

Despite this limitation, the article is interesting as it also provides a potential clinical impact of these findings proposing a practical algorithm.

Author Response

Reviewer #2

This article is a post-hoc analysis of a patient cohort. The authors have correlated NAFLD with Framingham Risk Score. Their results are original and interesting. The manuscript is clearly written in a straightforward way. 

Re: We thank Reviewer #2 for taking time to read our work and also for these encouraging comments.

The fact that none clinical endpoints but a surrogate risk score has been utilised should be addressed in the limitations section.

Re: This aspect is already mentioned in the limitation section of the manuscript. We have reworded the according sentence.

Despite this limitation, the article is interesting as it also provides a potential clinical impact of these findings proposing a practical algorithm.

Re: We thank Reviewer #2 also for the positive feedback on our proposed algorithm.

Reviewer 3 Report

In this manuscript, Niederseer et al perform a statistical analysis regarding NAFLD and risk for CV disease, using NFS and FNS.

The analysis is clear and presented well, and the results confirm other studies showing that NAFLD is present in a large percentage of the population, and that NAFLD is an independent risk factor for CVD. As in previous studies, the OR is fairly small. This study thus contributes nicely to an increasing pool of studies that define NAFLD as a “silent epidemic” as the authors write, and as a contributor to CVD.

I have a few suggestions for the authors to consider:

  1. Most important, since only 10 patients have F3-F4 grade fibrosis, the analysis of three groups of patients is weak. There are 138 patients with intermediate grade, and I suggest the merge the intermediate and high groups, or redefine their groups so that there are more patients in the “most sick” group.
  2. Since the effect of NAFLD - although significant - is small, the diagrams for treating patients are not warranted, and these recommendations are an over-interpretation of the data. This is true in particular for the 1% of patients with F3-F4 scores. 10 is too few to make such a claim. The diagrams also make the discussion too long and less readable.
  3. Another limitation of this study is the linearity of the models. This is implicit in the study, but should be stated, since the assumption of a linear contribution of so many related factors is a very rough estimation. In particular given the small effect of NAFLD on CVD.

Good luck

Author Response

Reviewer #3

In this manuscript, Niederseer et al perform a statistical analysis regarding NAFLD and risk for CV disease, using NFS and FNS.

The analysis is clear and presented well, and the results confirm other studies showing that NAFLD is present in a large percentage of the population, and that NAFLD is an independent risk factor for CVD. As in previous studies, the OR is fairly small. This study thus contributes nicely to an increasing pool of studies that define NAFLD as a “silent epidemic” as the authors write, and as a contributor to CVD.

 Re: We thank Reviewer #3 for his/her very kind comments.

I have a few suggestions for the authors to consider:

  1. Most important, since only 10 patients have F3-F4 grade fibrosis, the analysis of three groups of patients is weak. There are 138 patients with intermediate grade, and I suggest the merge the intermediate and high groups, or redefine their groups so that there are more patients in the “most sick” group.

Re: The grouping of patients follows logically in how the NAFLD fibrosis score works. It sorts subjects into three categories: F1-2 (rule-out fibrosis), F3-4 (rule-in fibrosis) and an indeterminant or intermediate group (neither rule in nor rule out). We have also considered the exact same strategy that Reviewer #3 suggests driven by the low number of rule-in. Because the NFS is designed like this we would rather leave the analysis like they are. But because of the valid point this reviewer is making, we ran secondary analyses in grouping F1-2 vs. the rest of the population (population at risk of fibrosis). We report these further analyses in Table 4. These findings essentially support the previous results.

  1. Since the effect of NAFLD - although significant - is small, the diagrams for treating patients are not warranted, and these recommendations are an over-interpretation of the data. This is true in particular for the 1% of patients with F3-F4 scores. 10 is too few to make such a claim. The diagrams also make the discussion too long and less readable.

Re: We agree with Reviewer #3 that the absolute numbers are small. However, we here report that approximately 1% of the general population of a colorectal cancer screening cohort in Austria has advanced fibrosis due to NAFLD. If one calculates the numbers for the entire country, there is huge number of unknown hepatic fibrosis due to NAFLD. As reviewer #2 explicitly mentions the value of our proposed recommendations we ask the editor to make a decision to either

  • Delete our recommendation
  • Transfer them to the supplementary material
  • Leave them as they are.
  1. Another limitation of this study is the linearity of the models. This is implicit in the study, but should be stated, since the assumption of a linear contribution of so many related factors is a very rough estimation. In particular given the small effect of NAFLD on CVD.

 Re: We agree with Reviewer #3 and thank him/her for this valid point. We therefore added a sentence on this limitation to the limitation section of the manuscript.

Good luck

Re: Thank you.